# Current Concepts and Methods in Tissue Interface Scaffold Fabrication

**DOI:** 10.3390/biomimetics7040151

**Published:** 2022-10-04

**Authors:** Oraya Vesvoranan, Amritha Anup, Katherine R. Hixon

**Affiliations:** 1Geisel School of Medicine, Dartmouth College, Hanover, NH 03755, USA; 2Thayer School of Engineering, Dartmouth College, Hanover, NH 03755, USA

**Keywords:** tissue engineering, regenerative medicine, scaffolds, tissue interface, 3D-printing, electrospinning, cryogels, hydrogels, solvent casting, gas foaming

## Abstract

Damage caused by disease or trauma often leads to multi-tissue damage which is both painful and expensive for the patient. Despite the common occurrence of such injuries, reconstruction can be incredibly challenging and often may focus on a single tissue, which has been damaged to a greater extent, rather than the environment as a whole. Tissue engineering offers an approach to encourage repair, replacement, and regeneration using scaffolds, biomaterials and bioactive factors. However, there are many advantages to creating a combined scaffold fabrication method approach that incorporates the treatment and regeneration of multiple tissue types simultaneously. This review provides a guide to combining multiple tissue-engineered scaffold fabrication methods to span several tissue types concurrently. Briefly, a background in the healing and composition of typical tissues targeted in scaffold fabrication is provided. Then, common tissue-engineered scaffold fabrication methods are highlighted, specifically focusing on porosity, mechanical integrity, and practicality for clinical application. Finally, an overview of commonly used scaffold biomaterials and additives is provided, and current research in combining multiple scaffold fabrication techniques is discussed. Overall, this review will serve to bridge the critical gap in knowledge pertaining to combining different fabrication methods for tissue regeneration without disrupting structural integrity and biomaterial properties.

## 1. Introduction

Damage caused by disease, congenital defects, and trauma results in an ongoing need to regenerate tissue; yet the human body alone remains inefficient in this task. The field of tissue engineering has evolved to combine three pillars (biomaterial scaffolds, cells, and bioactive factors) for targeting the growth and regeneration of new tissue. In a recent literature review, a search of the keywords “tissue engineering” and “regenerative medicine” resulted in 1.2 and 0.6 million results, respectively, on Google Scholar [1]. However, despite the rapid growth of tissue engineering research and the advantages both clinically and financially, there remain many gaps in knowledge that are preventing its translation from bench to bedside. Specifically, tissue-engineered scaffolds are typically fabricated to target one specific tissue that must be repaired, replaced, or regenerated. Rarely, however, do these tissues function independently, as the body is a dynamic and complex system where the interface between tissues—and resulting cross-talk—has been shown to guide growth, development, and regeneration [2]. Further, as traumatic injuries or congenital defects usually affect more than one tissue, treatment plans must reflect this.

When fabricating a tissue-engineered scaffold, it is imperative to closely relate the scaffold porosity, mechanical integrity, material choice, degradation, and cellular compatibility to the native tissue. Overall, previous research has demonstrated that pore sizes greater than 100 µm are appropriate for encouraging cell attachment, migration, and infiltration [3]. However, dependent on the tissue in question, these values may vary to encourage appropriate differentiation of stem cells. This desired pore size also extends to tissue survival, where the scaffold must support angiogenesis, or blood vessel formation, to ensure sufficient supply of nutrients, oxygen, and waste transport [4]. Additionally, the extracellular matrix (ECM) is analogous to the scaffold; thus, careful consideration of the polymer or combination of polymers is imperative for appropriate healing and cellular compatibility to occur [5,6]. Related to material, an ideal scaffold for any tissue type replicates the native mechanical integrity or encourages new matrix formation to achieve these mechanical properties following implantation [4]. Finally, degradation and bio-resorption are also key components to consider when choosing a scaffold material. The scaffold must be capable of degrading in a controlled manner such that new tissue can infiltrate the matrix at the same rate as scaffold degradation, with no toxic byproduct release [7]. By choosing the best combination of these various characteristics, an ideal tissue-engineered scaffold for bone, cartilage, ligament, muscle, skin, and tendon applications can be fabricated.

This review provides an overview of current knowledge of tissue interface. In order to progress tissue engineering to span multiple tissue types simultaneously, an understanding of the individual tissue components, properties, and healing capacity must be defined. Additionally, current tissue scaffold fabrication methods and material types must be explored. Together, these components provide the groundwork for fabricating scaffolds spanning tissue interface, where recent studies have begun to target the integration of multiple fabrication methods for this purpose. This review provides an overview of commonly targeted tissues in tissue engineering, specifically focused on their wound healing capacity and physical properties. Next, with these tissues in mind, commonly utilized scaffold fabrication methods are detailed to offer a background on available technology for creating these tissues. Finally, well-characterized scaffold material and additive information is listed to tie these sections together. The review then dives into current and future techniques to combine multiple tissue fabrication methods simultaneously, serving as a guide to directly target tissue interface.

## 2. Tissue Types

To appropriately target the regeneration of tissue interface, the structure and overall healing process of these tissues must first be individually understood. Bone, cartilage, ligament, muscle, skin, and tendon represent some of the most commonly targeted tissues in the tissue engineering and regenerative medicine field [6]. To appropriately target their relationship with one another, each tissue’s individual composition, function, and wound healing capacity should first be expanded upon.

### 2.1. Bone

Bone is a dynamic tissue that continuously undergoes remodeling throughout an individual’s lifetime. It consists of characteristic cell types forming a bone remodeling units (BRUs; i.e., osteoblasts and osteoclasts) surrounded by an ECM which is composed of both organic (organic type I collagen framework) and inorganic materials [8]. These inorganic minerals (i.e., calcium phosphate crystals) are embedded within and between the collagen fibrils where both the combination and overall arrangement of these structural components plays an essential role in bone mechanics [9]. Approximately 3–4 million BRUs are initiated each year, with 1 million actively engaged in bone turnover at any one time [10]. Osteoclasts and osteoblasts work successively in the same BRU to complete bone remodeling, which can be broken up into four phases [11]: First, osteoclasts are recruited in the initiation/activation phase. Next comes the resorption phase, where the osteoclasts start to resorb bone while MSC/osteoprogenitors are simultaneously recruited. The reversal phase follows, where osteoclasts undergo apoptosis and osteoblasts are recruited. Finally, the formation phase occurs where osteoblasts lay down a new bone matrix that subsequently mineralizes, resulting in complete bone remodeling [12]. After peak bone mass is reached, bone remodeling remains balanced for a few decades until age-related bone loss begins [13]. Note that bone-related disorders account for more than half of the chronic diseases in humans over 50 years of age [14].

Despite bone tissues natural ability to regenerate, complications from degenerative disease, congenital defect, traumatic injury, and bone tumor resection may result in an inability to heal, thus requiring surgical intervention. The current gold standard of treatment utilizes an autograft where donor bone is harvested from the patient (e.g., iliac crest) [15]. Despite the ability of bone grafts to appropriately integrate with patient tissue, there is a high potential for infection, donor-site morbidity, and postoperative pain [16]. While bone tissue engineering offers an alternative method for treatment, the scaffold must be appropriately fabricated to replicate the composition, structure, and mechanical properties of native bone tissue [16].

In order for a tissue-engineered scaffold to appropriately facilitate new bone growth, it should be both osteoconductive by supporting cellular adhesion, proliferation, and ECM formation on the scaffold surface/throughout the pores, as well as osteoinductive by inducing new bone formation through biomolecular signaling and recruitment of cells. Further, to achieve appropriate bioactivity, a diameter of at least 100 µm is necessary for the diffusion of essential nutrients and oxygen throughout the scaffold structure, as previously defined [17]. While it has been shown that pore sizes ranging from 200 to 350 µm are optimum for bone tissue to grow [18], another recent study demonstrated that multi-scale porous scaffolds (containing both micro- and macro-porosities) may further improve osteoconduction [19]. Despite the necessity of degradation rate matching new bone formation, this degradation rate varies with bone application, ranging from 3 to 6 months for cranio-maxillofacial application compared to 9 months in spinal fusion [4]. Finally, to directly match native bone biomechanics, the desired Young’s modulus will vary greatly based on cortical (15–20 GPa) or cancellous (0.1–2 GPa) bone type. Similarly, compressive strength also varies between cortical (100–200 MPa) and cancellous (2–20 MPa) bone [4]. However, studies have suggested that providing a matrix that supports initial bone healing (granulation tissue and hyaline cartilage) and thus, a lower modulus of 0.45–0.8 MPa, is acceptable to initiate bone healing [20,21,22,23].

### 2.2. Cartilage

Cartilage tissue is produced by chondrocytes which reside within the lacunae. The cartilage matrix is composed of fibrous tissue, as well as combinations of collagen, proteoglycans, and glycosaminoglycan [24]. There are several types of cartilage found in the body, and their structure and function are dependent on composition variations. Specifically, hyaline cartilage, the most common type of cartilage in the human body and found in the trachea, sternum, epiphyseal plate, and ventral segments of ribs, is primarily composed of type II collagen and proteoglycan [24]. It creates a surface with minimal friction and, thus, the ability to resist compressive force at the site of bone articulation [25]. Similar to hyaline cartilage, elastic cartilage contains type II collagen fibers; however, it also contains elastic fiber, which provides strength and elasticity to body parts such as ears, epiglottis, and eustachian tubes [24]. Finally, fibrocartilage contains type I collagen and significantly less proteoglycan than both hyaline and elastic cartilages. Therefore, it can resist higher degrees of tension/compression and is found in tendons, ligaments, intervertebral discs, and menisci [24]. Cartilage tissue lacks lymphatic and blood supply, instead relying on diffusion to and from adjacent tissue for nutrient and waste exchange. Similar to bone, cartilage is surrounded by a perichondrium-like fibrous membrane; however, this membrane does not support the regeneration of cartilage [26]. Thus, cartilage has a limited ability to repair itself following injury from wear, tear, and microfracture. Due to this, clinical pathology including osteoarthritis, traumatic rupture/detachment, costochondritis, and neoplasm continue to pose a challenge to physicians worldwide [26].

Current treatment options for cartilage defects include Pridie perforations and autologous osteochondral transplantation, but both possess inherent disadvantages [27]. In Peridie perforation procedures, microfractures are directly drilled through damaged cartilage areas, into the subchondral bone marrow space, to stimulate bone marrow for tissue repair. However, this method often results in fibrocartilaginous scar tissue with biomechanical properties inferior to that of hyaline cartilage. Additionally, the newly formed tissue does not prevent or slow the progression of degenerative arthropathy, is only applicable in small lesions (<4 mm), and carries the risk of thermal osteonecrosis [28]. Comparatively, in autologous osteochondral transplantation, hyaline articular cartilage is harvested from the host and transplanted into the defect area. While this procedure guarantees the transfer of a viable osteochondral unit, this rarely results in full repair due to the limited blood supply and, thus, healing capacity [28]. Additionally, this procedure carries a risk for donor site morbidity which may result in pain, instability, infection, and nerve injury [29]. Due to these limitations, tissue engineering offers a promising alternative method for inducing tissue repair and regeneration.

An ideal cartilage scaffold should possess sufficient porosity for cell ingrowth, along with appropriate biocompatibility and biodegradability [27]. Previous work has demonstrated that chondrocytes prefer a scaffold pore size between 250 and 500 µm for proliferation and ECM secretion. Specifically, it was shown that as pore size was increased from 50 to 500 µm, the rate of cell growth, amount of glycosaminoglycan secretion, and expression of gene markers for aggrecan, collagen type I, II, and X also increased [30]. Additionally, biomimetic architecture for cell adhesion, as well as sufficient mechanical strength to maintain both the shape and load bearing function at the site of implantation, is also imperative in cartilage tissue engineering. The aggregate modulus of cartilage is typically in the range of 0.5–0.9 MPa where the higher this value is, the less the tissue deforms under a given load. This translates to a Young’s modulus of 0.45 to 0.80 MPa [23].

### 2.3. Ligament

Ligaments are viscoelastic connective tissue with a highly organized composition that connect bones to joints, supporting the skeletal structure. They are composed of cells (i.e., fibroblasts), collagen (types I, III, and V), elastin, proteoglycans, and water. Due to poor vascularization, ligaments, similar to cartilage, have limited healing capability. Therefore, one of the most common ligament injuries is an anterior cruciate ligament (ACL) rupture, where approximately 100,000–200,000 ACL ruptures occur each year in the United States [31]. As the ACL is one of the major intra-articular ligaments of the knee and plays a critical role in body kinematics/stability, restoring the structure and function post-injury is essential [32]. Further, if the ligament is not properly repaired, the loading of the joints can lead to abnormal stress on the articular cartilage and result in an early onset of osteoarthritis [33].

The current gold standard treatment for ACL rupture is surgical tissue reconstruction [34]. This surgical procedure involves reconstruction of the ligament in the center of the knee using either host (autograft) or donor (allograft) tissue. Based on a systematic review, this reconstruction method has better functional outcomes than rehabilitation alone [34]. However, despite autograft success rates, limited availability and donor site morbidity, including anterior knee pain, tendonitis, patellar fracture, and muscle weakness, remain prevalent concerns [35]. In comparison, an allograft carries the risk for transmission of blood-borne disease and delayed biological incorporation [36]. Thus, tissue-engineered ligaments could provide a viable alternative treatment method to restore tissue function.

An ideal scaffold targeting ligament regeneration should behave similarly to natural ACL and provide sufficient mechanical strength. The two primary cell types used in ACL regeneration are mesenchymal stem cells (MSCs) and ACL fibroblasts due to their ability to differentiate into various lineages and their role as native cells in the ligament tissue, respectively [37,38]. Previous work has shown that a minimum pore diameter of 200–250 µm encourages soft tissue ingrowth and capillary supply [39]. Mechanically, ligaments are complex structures that display triphasic behavior when exposed to strain: the non-linear, linear, and yield/failure region. The non-linear or toe region is where the ligament exhibits a low amount of stress per unit strain. This region is then followed by the linear region, where there is a noted increase in stress per unit strain. Lastly, the yield/failure region is a region that shows a slight decrease in stress per unit strain and marks the failure of the ligament. This behavior is due to composition and arrangement within the tissue and should be replicated in the tissue engineered scaffold [32]. The average ACL can withstand cyclic loads of approximately 300 N, 1–2 million times per year. The established standards for ACL grafts are 1730 N for the tensile strength [40], 182 N/mm for linear stiffness [40], and 12.8 for energy absorbed failure [41]. Additionally, the scaffold should be biocompatible, biodegradable, and cause minimal/no inflammatory response. Lastly, the tissue-engineered ligament should degrade at a controlled rate, allowing new tissue to receive the appropriate load level without the risk of rupture [32]. There are multiple fabrication parameters that can be manipulated to achieve this, including morphology, molecular weight, and degree of crystallinity [42].

### 2.4. Muscle

Muscle tissue comprises approximately 45% of the human body mass and is capable of some degree of regeneration [43]. Muscle ranges from striated (i.e., cardiac and skeletal) to smooth muscle. Cardiac muscle cells are non-fatiguing and produce the coordinated, rhythmic pulse to pump blood. Comparatively, skeletal muscle is voluntarily controlled where the cells have high energy requirements and become fatigued after long periods of use. Finally, smooth muscle is found in many areas of the body, including the lining of organs [44,45]. A key component of skeletal striated muscle is its organization into fibrous bundles, or myofibers, where each myofiber is a muscle cell. In addition to muscle cells, there is robust vasculature that aids in delivering nutrients to these cells [44]. The majority of the 150,000 open fractures in the United States include soft tissue loss, and out of the severe open tibial fractures, 58% involve muscle damage [46]. Following injury, damaged myofibers and inflammatory cells signal neutrophils and macrophages to the site of injury. Muscle stem cells (i.e., satellite cells) located between the basal lamina and sarcolemma of the muscle fibers are activated and have the capacity to proliferate and differentiate into myoblasts. Following nerve formation, the myoblasts fuse to form new microfibers. Despite this well-defined pathway, tissue regeneration is often unsuccessful following traumatic injuries, volumetric muscle loss, tumor ablation, and denervation. In such injuries, the muscle cannot be regenerated and thus, scar tissue forms in the area of injury. Scar tissue serves as a placeholder for the lost muscle, but in many cases causes more damage due to its inferior mechanical properties [43].

Current treatment options for excessive muscle loss include cell therapy and muscle grafting, more specifically free-flap transfer and extensive physiotherapy [46]. Cell therapy involves the injection of muscle cells cultured outside the body, whereas a muscle graft is obtained from a second surgical site and implanted at the site of injury. Both techniques have limitations including inappropriate innervation of the implanted muscle and donor site morbidity [43]. Tissue engineering aims to improve the muscle healing capacity while mitigating these risks of donor pain and insufficient integration.

Muscle is primarily composed of mature fibers, or myofibers [47], derived from satellite stem cells. Previous work has demonstrated that satellite cells are influenced by their surrounding cellular microenvironment. Specifically, ECM proteins such as collagen IV have been shown to influence the self-renewal capacity of satellite cells [48]. Other regulatory proteins troponin C, I, T, and tropomyosin all assist in muscle contraction [47]. Tissue engineering of muscle should target pore sizes of 50–160 µm to induce regeneration at the injury site [49]. The ECM plays a large role in the development and function of the anisotropic muscle tissue and, therefore, it is important to consider how to best replicate the ECM when assembling tissue-engineered scaffolds for muscle regeneration [46]. This includes mechanical properties such as a 12 kPa elastic modulus and in vitro strain measurements of 1.8 ± 0.67 kPa [50]. It is also important to consider the increases in the stiffness as patient ages [51]. Tissue engineering offers the potential to create an optimal environment for muscle regrowth by providing chemical cues and physical stimulation to support muscle cells in regenerating the tissue [43].

### 2.5. Skin

Skin is the largest organ in the body and consists of a range of properties and structural features that allow it to handle a variety of mechanical forces. Skin tissue can be broken up into three primary layers: the epidermis, dermis, and hypodermis. The outermost layer and external covering of the epidermis, or the stratum corneum, is waterproof and inhibits foreign body entry. Epidermal keratinocytes are the primary cell within the epidermis and contribute to wound repair [52]. Other epidermal cells include melanocytes, Langerhans, and Merkel cells [53]. The dermis is a thick, connective tissue layer that is composed of fibroblasts, sweat and sebaceous glands, blood vessels, and the ECM. The dermis is highly populated by fibroblasts, which are cells that produce ECM components including collagens, fibronectin, and laminins [54]. Finally, the hypodermis consists of adipose tissue cells, and provides support between the skin and other skeletal structures [53]. The wound healing of skin is a precisely tuned process that can be broken into four general steps [53]: First, hemostasis occurs where vasoconstriction is driven by signals released from cells neighboring the injury site. Following this, inflammatory cells (i.e., neutrophils and macrophages) are triggered to infiltrate the area by chemical factors associated with tissue damage. Additionally, keratinocyte regeneration occurs and new vessels are formed. In the proliferation phase, angiogenesis is carried out by epithelial cells and genes that are responsible for fibroblast proliferation are transcribed/translated. Finally, remodeling results in the regeneration of the injured epidermal and dermal layers [53]. For surface-level and minor injuries, the skin is capable of following this pathway to achieve healing, but scarring often occurs. For deeper injuries caused by burns, traumatic injury, etc. [55], the body is incapable of adequate healing and external clinical support is required. If this healing assistance is not received in an appropriate time frame, chronic wounds, fluid and electrolyte imbalance, and even sepsis can result, which in turn can lead to death [53].

The current gold standard for treating skin injury is autologous split-thickness skin grafting (STSG), which involves extracting grafts from the thighs or back for implantation at large wound sites [56,57]. Despite the common use of skin grafting, associated complications may occur including dehiscence, necrosis, infection, seroma/hematoma, and inappropriate vascularization [58]. Additionally, success relies heavily on patient compliance, as well as an interprofessional team to manage patient care both before and after surgery [59]. Therefore, there are many advantages to targeting novel technologies for creating tissue substitutes to reduce the grafting, while obtaining appropriate skin healing and regeneration. The field of tissue engineering provides a practical avenue to fabricate skin substitutes for clinical applications to restore and regenerate skin tissue following injury.

To appropriately target regeneration and replicate the collagen fibrils found in native skin ECM, tissue-engineered fiber diameters must fall in the range of 50 to 500 nm, while maintaining mechanical integrity [60,61,62]. The basement membrane is the stiffest layer of the skin, sustaining 1–4000 kPa [63]. However, different areas of skin throughout the body possess variable mechanical requirements. A 2017 study [64] examined the mechanical properties of skin samples from five different areas of the human body targeted during nasal and ear reconstruction: forehead, forearm, temporoparietal, post-auricular and submandibular neck. It was demonstrated that the skin thickness (0.87–1.4 mm), Young’s modulus (1.03–1.28 MPa), and stress relaxation rate (4.74–7.8 MPa) varied between all of these locations [64]. Thus, implantation location should be considered, along with desired degradation rate with respect to the extent of skin tissue injury.

### 2.6. Tendons

Tendons are responsible for transferring forces from muscle to bone and are critical for joint mobility and stability [65]. Overall, tendons have a low capacity to heal, primarily due to a lack of blood supply. They are primarily composed of collagen and are highly viscoelastic, where tendon response to stress is nonlinear and time-dependent. Thus, as the strain rate varies, a tendon will exhibit different stress and strain values at failure [66]. Further, depending on the strain value that the tendon is exposed to, the overall structure will vary. For example, under a relaxed state, tendon fibers form crimped packings, whereas exposure to strain leads to fiber straightening; however, an applied strain exceeding 6% will result in tendon fiber rupture [67]. In addition to these mechanical properties, biochemicals also influence tendon function through mechanosensing. In one study conducted in rats, 64% of tenocytes, or tendon cells, contain a primary cilium that aids in sensing mechanical/chemical changes in the surrounding environment [68]. The orientation of the cilia varies greatly depending on the strain experienced by the tendon cells, indicating the cilia is critical to tendon function. Focal adhesion kinase (FAK) is another mechanosensor that plays a critical role in tendon cell differentiation. Finally, integrins are responsible for transducing signals to the cell cytoskeleton [67].

The current gold standard of treatment following tendon injury involves surgery which may include suturing the ends of the ruptured tissue. In more complex cases, inclusion of a tendon graft may be required to provide a base material to initiate repair [65]. These treatment options possess disadvantages including insufficient mechanical stability, as well as a risk of graft failure [69]. Thus, there remains a need to create tissue-engineered scaffolds to replicate tendon structure and restore function at the site of injury.

To encourage cellular growth, previous work has identified a pore size of 300 µm to be optimal for tendon stem cells [70]. Mechanically, the elastic stiffness of tendon is approximately 0.02 GPa; however, as tendons vary in function, as do their mechanical properties. For example, the Young’s modulus of a patellar tendon and tibialis anterior tendon are approximately 660 and 1200 MPa, respectively [71,72,73]. Age and disease of the patient can also have an impact on tendon mechanics, where the Young’s modulus of the patellar tendon decreases from the previously mentioned 660 MPa in donors under 50 years old to 504 MPa in donors over 64 years old [71]. Due to the mechanically responsive nature of tendons, it has been proposed that tissue engineering should utilize bioreactors to apply biaxial or uniaxial mechanical stimulation and appropriately support the growth of this tissue.

Following a detailed understanding of the most common tissues targeted in the field of tissue engineering, an overview of scaffold fabrication techniques opens avenues towards targeting the specific design of these different tissue types.

## 3. Scaffold Fabrication Techniques

Tissue engineering is comprised of a specific combination of biomaterial scaffold, cells, and bioactive factors to encourage regeneration at a site of injury or defect. As the scaffold serves as the structural ECM, appropriate fabrication is necessary to initiate tissue-specific healing. There are numerous methods to fabricate 3D scaffolds which can be directly tailored for the desired tissue; thus, scaffolding constructs are often adaptable to specific tissue size, orientation, and physiology [74,75,76,77]. However, modification of tissue-engineered scaffold geometry can have a negative impact on scaffold porosity, mechanical properties, degradation, and cell compatibility, thus rendering the structure useless in tissue imitation. Thus, selecting the appropriate tissue-engineered scaffold fabrication method is vital to ensuring proper tissue growth and regeneration [78,79].

### 3.1. 3D-Printing

Additive manufacturing, or 3-dimensional printing (3D-printing), is a common method for creating tissue-engineered scaffolds. Before 3D-printing, manufacturing was dominated by removal-based processes that created objects by cutting away at solid material. Alternatively, 3D-printing utilizes a bottom-up approach to produce an object through layer by layer ink deposition beginning at the model base [80,81]. The most common types of 3D-printing include binder 3D-printing, digital light processing (DLP), fused deposition modeling (FDM), inkjet, selective laser sintering (SLS), and stereolithography (SLA). To 3D-print in tissue engineering, a model of the desired scaffold is generated using a computer-aided design (CAD) modeling software. However, with the rising availability of technologies such as MRI, CT, and ultrasound, it has become possible to use clinical patient scans to develop patient-specific models [82]. Following the initial modeling, a software is then used to convert the CAD file into a surface tessellation format (STL) that breaks the model into triangles [83]. STL files contain x,y,z coordinates of triangle vertices and the vector normal to each triangle [80]; this format supports slicing the 3D object into 2D layers, providing the necessary information for production [84]. Scaffold properties can be tuned by adjusting many variables depending on the type of 3D-printing, including natural (e.g., chitin, collagen, cellulose, gelatin) or synthetic (e.g., polycaprolactone, polyglycolide, copolymers) ink type [80], temperature at which the ink is printed, print layer density, and extrusion rate [83]. Further, the mode of deposition can vary including light-based, powder, and extrusion-based 3D-printing.

SLA and DLP are two light-based 3D-printing modes (Figure 1A,B). SLA, the oldest form of 3D-printing still used today, employs a direct UV HeCd laser writing technique to harden plastic [85,86]. Benefits to SLA include material availability, good surface texture, and high resolution [83]. Comparatively, DLP uses a spatial light modulating element to produce a 2D light projection onto photosensitive ink [87]. This projection enables the formation of a whole 2D slice of the object at one time and is fast, but less accurate.

Binder 3D-printing, powder 3D-printing, or the “drop on powder technique”, involves the deposition of inkjet liquid printing binder solution onto a powder base. The binder solution then selectively joins the powder to form a 2D slice of the desired part. After the first layer is created, the platform is lowered and a fresh layer of powder is applied; this process is repeated until the full object is created and unbound powder is subsequently removed [88]. While the powdered base adds support to the 3D-printed object during fabrication, the printer nozzle is prone to clogging and this printing results in low resolution [80]. Comparatively, SLS (Figure 1D) uses a CO_2_ laser to heat-fuse powder particles together [89], similar to binder-based 3D-printing. With this technique, the intensity of the laser, scanning rate, and material properties are all variables of interest [83].

Extrusion-based 3D-printing encompasses both FDM (Figure 1C) and inkjet 3D-printing, where the former layers thermoplastic material through a temperature-controlled nozzle. Once the thermoplastic filament is melted and dispensed through the nozzle to create a single layer of the object to be formed, the platform is lowered before applying the next layer [90,91]. Variables of interest include thickness of each layer, rate of the filament, and plotting speed [83]. Inkjet 3D-printing works similarly to commercial paper printers in that droplets of ink are deposited to form each layer; however, the ink is crosslinked after each layer is deposited, permitting additional layers to be added. These printers may also possess multiple nozzles, allowing different types of materials to be incorporated into a single object [87]. 

One advantage of using 3D-printing for generating scaffolds is the ability to create a cost effective, porous scaffold with detailed control [80,81]. In addition, partnership with imaging systems supports patient personalization. It is anticipated that as imaging and printing capabilities advance, the resulting resolution will also improve to enable the printing of detailed and clinically accurate scaffolds [80]. Traditionally, additive manufacturing methods also produce less waste than typical manufacturing methods [82]. 3D-printing has been used for a wide variety of tissue engineering applications including bone, skin, cartilage, and vascular networks [80]. However, due to its ongoing, rapid development, the field lacks standardization and systematic execution methods [80].

### 3.2. Bioprinting

Bioprinting is an advanced form of 3D-printing where the ink (bioink) can be a mixture of both biomaterials and cells, or cells alone. Bioinks contain cell types that are desired in the final implant, and they can be utilized in scaffold-based or scaffold-free forms. Scaffold-based techniques involve bioinks with cells embedded in a matrix that can be printed directly onto a pre-formed biomaterial matrix [92]. Hydrogels (natural or synthetic) are most commonly used in scaffold-based processes, and are crosslinked to mimic the ECM. Scaffold-free bioprinting involves cells that are printed without an exogenous biomaterial and mimics embryonic development [93,94]. These bioinks consist of cell aggregates pre-made into geometrically relevant shapes (cylinder, toroids, torus, or honeycomb), that encourage cells to produce ECM components [92]. These cell aggregates can be found as several forms of inks, including tissue spheroids (cell aggregates 200–400 µm in diameter), cell pellets that can be transferred to a mold, and cylindrical tissue strands [93,94].

The three main bioprinting modalities are extrusion-based (EBB), droplet-based (DBB), and laser-based (LBB) bioprinting [93,94,95]. EBB consists of a nozzle that dispenses bioink continuously, and is used in scaffold-based and scaffold-free printing. The ink can be dispensed pneumatically, with a piston, or by screw (Figure 2B). DBB and LBB are both used for scaffold-based bioprinting. In DBB, bioink droplets can be stacked in order to create the 3D structure. Dispersion of ink is made possible via heating or pressure changes induced by a piezoelectric actuator (Figure 2A). LBB typically consists of light and photopolymerizable ink materials, but it can also print via laser induced jet formation [93,94]. In this set up, a laser is focused on the energy absorbing layer, leading to a vapor bubble that provides pressure to dispense bioink droplets (Figure 2C) [81]. Following printing using any of these methods, the construct is commonly placed in a bioreactor to optimize cell proliferation, tissue remodeling, and maturation prior to implantation [96,97].

Scaffold-based bioprinting is economical, scalable, and has high resolution, but cellular interactions are limited, regeneration time is long, and the completed scaffold can be toxic depending on the materials used. Scaffold-free methods are more cell-friendly and ensure rapid tissue maturation; however, it is costly, not scalable, requires a large number of cells, and may possess low mechanical integrity [92]. Further, in both scaffold-based and scaffold-free bioprinting, optimizing settings for nozzle diameter, extrusion pressure (if using EBB), and printhead speed for a specific application can be a challenge [98]. Overall, bioprinting has been successfully used for fabricating neural tissue [99], liver organoids that prolonged the survival of mice with liver failure [100], skin [101], and bone for treating osteochondral defects in an osteoarthritic rat model [102]. Another recent study introduced a new technique called bioprinting-assisted tissue emergence (BATE), and used this to demonstrate that organoid forming stem cells were capable of forming self-organizing tissue structures such as lumens, vasculature, and villi [98].

### 3.3. Electrospinning

Electrospinning is a polymer fabrication process capable of producing both microfibers and nanofibers. The traditional electrospinning setup includes a syringe containing a polymeric solution, a volumetric pump to regulate the solution extrusion rate, a power supply with a voltage range of several kV, and a grounded, or negatively charged, collector for fiber deposition [103]. When the electric potential is applied between the syringe and the collector plate, charges accumulate and focus on the surface of an emerging polymeric droplet at the needle tip. Due to the high voltage applied at the needle tip, the electric force field overcomes the cohesive force of the solution, dominated by surface tensions, and an electrically charged polymer jet is extruded. The jet moves toward the collector and is elongated by electrostatic interactions between neighboring jet segments. As the solvent of the solution evaporates, the jet solidifies into a fiber and collects in the form of a fibrous mesh (Figure 3). One of the limitations of electrospinning is its inability to produce large 3D scaffolds, where the typical product is a flat 200–500 µm thick sheet [104]. Creating a thicker fiber requires a large amount of polymer and time; thus, this renders the technique inefficient in creating scaffolds for large defects [104].

Over the past decade, electrospinning has gained popularity in tissue engineering due to its ability to create scaffolds that mimic the ECM structural organization, allowing for cellular adhesion and growth. Additionally, as both synthetic and natural materials can be used, it mitigates a negative immune response or disease transmission. Electrospinning has been used to target many tissue types, including bone, cartilage, ligament, and skin, and is both low-cost and easy to scale-up [103].

The final electrospun scaffold is affected by three main components: solution properties, processing parameters, and ambient parameters. Solution properties include volatility, polymer concentration, and polarity. For example, if the solution is too volatile and fluid jets are collected prior to full solvent evaporation, the fibers may be flattened upon impact with the collector’s surface or adhere to neighboring fibers. Additionally, if the deposited fibers settle on the collected fibers that are still fluid (i.e., before the solvent fully evaporates), then these fibers can merge to create a conglutinate network [106]. Processing parameters include applied voltage, flow rate of polymeric solutions, temperature, the distance between the tip and the collector (working distance), needle diameter, and nozzle configuration. These factors can all influence fiber morphology. For example, higher flow rate decreases fiber diameter and may result in spindle-shape beads becoming spherical. Specifically, Martino et al. [103] demonstrated that by manipulating the polymeric solution flow rate, electrospinning could create a PLLA fiber with the morphology of beads without fiber, fibers with beads, and fibers without beads. Lastly, the ambient parameters include temperature, humidity, pressure, and atmosphere.

### 3.4. Gas Foaming

Gas foaming is completed by administering gas into a liquid biopolymer at high pressures and room temperature over a period of time (typically around 3 days) [107]. When the gas is solubilized into the polymer, the polymer-gas combo is cooled. Once uniform temperature throughout the object is achieved, the pressure is dropped to standard atmospheric conditions, resulting in an end product that is a stable, porous scaffold [107]. The high-pressure component of this method may limit its application with materials that are pre-dosed with cells or other bioactive materials. However, gas foaming results in high porosity, promoting cell attachment, nutrient and waste transfer, and gas diffusion [108]. Note that these characteristics are heavily dependent on pore size, where while large pores allow for improved nutrient and waste exchange, cell attachment may be inhibited (and vice versa for small pores) [108].

To prepare a gas foaming scaffold, an aqueous solution containing both a biopolymer and a surfactant is molded into the shape of the scaffold [109]. Then, gases, or chemical/physical blowing agents, are applied and spread throughout the polymer phase [110]. Common chemical blowing agents include reactants that come together or decompose into gasses such as CO_2_ or N_2_ [107,110,111]. Under chemical blowing, scaffolds are limited to hydrophilic materials as the gas is hydrophobic and polarity is necessary for generating the porous scaffold. However, the equipment for this technique is generally available and easily scaled-up for mass production [110]. Despite this, the reactions often involve side product salts and it is not as easy to ‘dose’ the amount of gas introduced. In comparison, physical blowing agents solve both of these challenges by using agents that are typically volatile liquids which expand foam via evaporation (Figure 4) [110]. A reactor is used to administer a set amount of gas/volatile liquid through a glass capillary into the aqueous biopolymer solution, which permits some control over scaffold porosity [110]. Foaming can also be done in a microfluidic fashion. Microfluidic technologies enable fluid flow control at the microscale, which allows for fine tuning of a uniform porous environment. This added uniformity improves the ability to distribute chemical stimuli between cells throughout the scaffold [110].

Porosity and pore size ranges obtainable via gas foaming depend on the material used. Studies have demonstrated a large variation in sizing including: alginate hydrogels with pore sizes ranging between 100–300 µm macropores and 30–80 µm micropores [113]; gelatin with pore sizes ranging between 230 µm for macropores and 90 µm for micropores [113]; PCL foams with porosity of 78–93% and pore sizes ranging from 10–90 µm [114]; and PVE with pores sizes ranging from 3–15 µm. Generally, gas foaming fabrication can obtain porosity of up to 93% and pore sizes of 100 µm [111]. While some studies have shown the ability to fine tune the porous network with this fabrication method [111], others have demonstrated difficulty in strictly controlling these properties [107,115]; this may vary based on material type. Another benefit of gas foaming is that it does not require the use of organic solvents that could be harmful for cells and tissues [111].

### 3.5. Hydrogels

Hydrogels are 3D networks of hydrophilic polymers, crosslinked through covalent bonds and other physical intramolecular or intermolecular attractions [116]. Due to the presence of hydrophilic moieties along the backbone chain (e.g., carbonyl, amide, amino, and hydroxyl groups), hydrogels can absorb up to several thousand percent of their weight in water or other fluids. Note that this process can occur quite rapidly without related gel dissolution [116]. Hydrogels can be prepared in multiple ways depending on the desired structure and application, including natural, synthetic, or semi-synthetic polymers. Following polymer choice, methods of preparation span free radical polymerization, as well as irradiation, chemical, or physical crosslinking [116].

Free radical polymerization is the preferred method for monomer- and natural-based hydrogels [117], where the latter requires functional groups that are radically polymerizable [118,119]. Conveniently, free radical polymerization can be performed in either solution or as bulk. Solution polymerization is preferable for large quantities of hydrogel synthesis, where the most common solvent is water; however, this type of polymerization requires a solvent removal step, which can be time-consuming [116]. In comparison, bulk polymerization does not require a solvent and is thus, the preferred method for rapid fabrication [120]. Irradiation crosslinking creates a reactive site along the polymer strands where radical sites are subsequently combined; this method is effective, especially when combined with a simultaneous sterilization process [116]. Further, unlike free radical polymerization, this method does not require catalysts or additives for the initiation reaction and can produce hydrogels as either solution or as bulk [116]. Despite the control over crosslinking, solution polymerization is preferred as it requires less energy for macroradical formations and reduces the mixture viscosity [121]. In chemical crosslinking, a bifunctional crosslinking agent is added to a diluted solution of hydrophilic polymers where both natural and synthetic hydrophilic polymers can be used [116,122]. Finally, crosslinking can be achieved through physical crosslinking using polyelectrolyte complexation, hydrogen bonding, and hydrophobic association [116]. In polyelectrolyte complexation, links are formed between pairs of charged sites along the polymer backbone where the pH of the system determines the electrolytic link stability [123]. In hydrogen bonding, the bond is formed through the association between an electron-deficient hydrogen atom and a functional group with high electronegativity. Many factors affect the hydrogel characteristics formed by this method including polymer concentration, molar ratio, and solvent type [123]. In hydrophobic association, the hydrophobic micro-domains act as an associate crosslinking point in the entire polymer structure, surrounded by the hydrophilic water-absorbing regions [116]. This method is low-cost; however, poor interfacial adhesion causes the resulting hydrogel to possess poor mechanical properties [116].

Given their similarity to macromolecular-based components in the human body, hydrogels have excellent potential as scaffolds in tissue engineering [124]. Further, they offer mechanical properties similar to native connective tissue which supports both cellular adhesion or suspension within the 3D network. Note that if cellular adhesion is preferred, the hydrogel must include appropriate peptide moieties on the surface or throughout the bulk of the scaffold to support attachment [116]. For example, the RGD (arginine-glycine-aspartic acid) adhesion sequence is commonly incorporated to improve cellular migration, proliferation, growth, and organization [125]. Furthermore, the surface characteristics can be easily manipulated with the addition of adhesive bio-moieties to facilitate interactions between the hydrogel and surrounding environment [116]. Additionally, hydrogels possess high biocompatibility and biodegradability. While natural polymer-derived hydrogels typically provide adequate biocompatibility, synthetic polymers may elicit a significant immune response and, therefore, may require purification [124]. In terms of biodegradation, studies have shown that cells seeded in proteolytically degradable hydrogel scaffolds have a higher proliferation rate and ECM production in comparison to non-degradable hydrogel scaffolds [126]. The addition of growth factors or endothelial cells can assist in stimulating vascular formation from surrounding tissue. Figure 5 illustrates vascular formation within a hydrogel [116]. Finally, hydrogels can also be classified as either durable (e.g., polyacrylate-based hydrogel) or biodegradable (e.g., polysaccharides-based hydrogen), providing information on their mechanics [116]. This is dependent on the hydrogel stability in the physiological environment. Further, based on the hydrogel response to environmental stimuli, they can be classified as either smart or conventional. While these both have similar methods of preparation and characterization, the smart hydrogel can modify swelling behavior, network structure, and/or mechanical characteristics in response to environmental stimuli [116]. These stimuli may include pH, temperature, light, ionic strength, and electric field [127,128,129].

### 3.6. Cryogels

Cryogel fabrication closing mimics that of hydrogels, but creates constructs below the melting point of a solvent. While hydrogels can mimic the biological properties of ECM, they may possess insufficient vascularization due to inadequate pore sizing to support angiogenesis throughout the scaffold [130,131,132]. Cell viability is dependent on close proximity (within a few hundred µm) to blood vessels; thus, limited vasculature commonly leads to decreased cell viability in hydrogel scaffolds [116]. Comparatively, cryogels possess macroporous structures resulting in improved vascularity, rapid swelling, and enhanced cellular infiltration [17].

For cryogel fabrication, gel precursors are first crosslinked, either physically or chemically, and then subsequently frozen. Cryogel synthesis primarily occurs in the frozen state, including the chemical reaction leading to gelation [133,134,135,136]. The temperature for freezing typically ranges between −5 and −20 °C, as this is the common range for solvent crystallization. The resulting solvent crystals act as porogens, surrounded by the hydrogel constituents which remain in liquid micro-phase [135,137,138]. Following freezing, the cryogel is thawed at room temperature, where the ice crystals melt out, leaving an interconnected, macroporous structure [134,135,137,138,139,140,141]. Note that the pores become rounded upon hydration due to the liquid and pore wall surface tension. An overview of the cryogel fabrication process is shown in Figure 6.

There are a number of variables that determine the physical properties of cryogels, including type of crosslinking, polymer concentration, temperature of gelation, cryo-concentration, and cooling rate [134,142]. In terms of crosslinking, cryogels can form through physical or chemical crosslinking using natural and synthetic polymers [137]. Note that physical crosslinking has been shown to generate scaffolds with resultant pore sizes of less than 10 µm, which is considered too small for tissue engineering applications [137]. In comparison, chemical crosslinking often allows for large pore size formation, typically on the order of 100–200 µm [137]. However, chemical crosslinking can carry the risk of introducing an unreacted chemical crosslinker which may reduce cell viability [143]. Regarding polymer content, both the percentage and the molecular weight of the polymer affect the final pore size. Compared to gel solutions with higher molecular weight, lower molecular weight solutions at equal mass concentration have been shown to create scaffolds with larger pore size [144]. Similarly, higher concentrations of polymer increase the stiffness of cryogel, but generate relatively smaller pore size when compared to those with lower concentration [87]. Higher concentrations of polymers increase the availability of crosslink groups, but at the same time decrease the availability of free water [137]. For temperature, studies have shown that lower cryogelation temperatures generate smaller pore size [145], as the solvent crystallizes more rapidly at lower temperature, resulting in a greater number of smaller solvent crystal nucleation [146]. Further, numerous studies have demonstrated a linear relationship between temperature and pore size [146], as well as an optimal cryogelation temperature for maximizing pore size [147]. Lastly, cooling rate can also affect the pore size of cryogel. Hydrogen bonds within the structure lower the freezing point of the solution, but the solution must be partially crystalized to form the solidified solvent porogen [137]. Therefore, the rate of crosslinking must be slower than the rate of solvent crystallization in order to create a homogenous macro-porous structure [148,149,150]. If the rate of crosslinking is faster than the rate of solvent crystallization, polymerization will occur without the solvent porogens, creating non-macroporous gels. By slowing the rate of crosslinking, there is an increase in crystal formation and growth [149,151].

### 3.7. Solvent Casting and Particulate Leaching

Solvent casting and particulate leaching (SCPL) is a fabrication process used to create thin polymeric films from solution [152]. First invented in 1999 to overcome the uncontrolled pore size and porosity seen with fiber bonding [153], it is considered a reliable and controllable technique for developing biodegradable porous scaffolds with uniform pores [154]. The primary advantage of this process is the ability to control the thickness and distribution of the pores throughout the film [152]; however, this method can only produce scaffolds in a thin film layer with limited mechanical properties and carries the potential for residual salt particulates [155]. 

SCLP involves the casting of an organic polymer solution, containing a crosslinker and salt particulates (porogens), into a mold [154]. Porogen agents commonly used include sodium chloride [156,157], ammonium bicarbonate [158], and glucose [159], where different porogen material yields varying crystal sizes. For fabrication, first the polymer is dissolved in an organic solvent. This solution is then combined with a porogen and transferred to the mold, such as a Petri dish. Once the solution solidifies, the solvent is removed from the solution via evaporation or lyophilization, leaving behind the porogen within the polymer structure. The porogen is then “leached” out by immersion in an aqueous bath which dissolves the particles/salt within the matrix [160]. Figure 7 provides an overview of this process.

The final properties of the scaffold depend on various processing parameters. For example, the pore size can be controlled by varying particle/polymer ratio and porogen size [162]. Previous work has demonstrated that the SCPL method allows for fabrication of scaffolds with pore sizes ranging between 30 and 300 µm and a porosity of 20–50% [162]. Further, the polymer and solvent components, as well as their ratio, can also be modified to vary the final properties of the scaffold [152]. Additional parameters to modify the resulting scaffold include the casting substrate, casting temperature, and drying conditions (i.e., indirect heating, heating by radiation, and air drying) [163].

By understanding tissue composition and the available fabrication methods, the material composition and potential additives to enhance scaffold properties can be explored.

## 4. Materials and Additives

Choosing the biomaterial for a scaffold is an important component in a tissue-engineered scaffold design as it must closely replicate the native ECM. Natural, synthetic, and ECM-derived materials have previously been targeted for all previously mentioned fabrication techniques [164]. Natural polymers typically possess bioactivity, biocompatibility, non-toxic biodegradation byproducts, and cellular interaction sites; however, these materials also may possess disadvantages including poor mechanical strength, reduced tunability, and potential contamination. Comparatively, synthetic polymers allow for tunability and bulk synthesis, but lack cell adhesion sites [165]. Finally, ECM-derived polymers provide cues and structure similar to the native environment, but lack mechanical strength. Thus, by noting the advantages and disadvantages of these material types, the appropriate polymer, or combination of polymers, can be chosen for the desired tissue application. An overview of commonly targeted polymers in tissue engineering is provided in Table 1.

The primary goal of tissue-engineered scaffolds is to mimic the function of native ECM in a targeted human tissue. This scaffold should provide mechanical stability and structural support for cellular attachment, infiltration, and differentiation [280]. The scaffold may also be required to serve as a source of growth factors, providing a space for both vascularization and new tissue formation [280]. Overall, natural collagen-based scaffolds have previously demonstrated high porosity and permeability; however, these scaffolds alone experience rapid degradation within the body and possess poor mechanical properties [281]. Similarly, synthetic biodegradable polymers also possess disadvantages due to a lack of bioactivity [282]. Therefore, it is advantageous to incorporate biological additives to jumpstart healing at the site of scaffold implantation.

Previous work has incorporated bioceramics into scaffolds to improve toughness and stability [283,284]. These components have been added as fillers or coatings to polymer matrices to create a composite scaffold with improved strength and stiffness, as well as enhanced bioactivity [285]. Further, these components include ions commonly found in physiological environments (i.e., Ca+2, K+, and Na+) [286]. Overall, bioceramics can be separated into three groups: (i) inert (e.g., alumina and zirconial), (ii) bioactive (e.g., bioactive glass), and (iii) resorbable ceramics (e.g., a-tricalcium phosphate (TCP) [287]. One of the most commonly used bioactive bioceramics in tissue scaffolding is hydroxyapatite (HA). HA is an inorganic calcium phosphate molecule that resembles native biological apatite from the mineral phase of calcified tissue in the body [286]. Thus, HA is commonly targeted in bone tissue scaffolding due to its osteoconductive nature, promoting bone growth on the scaffold surface and within the pores [288]. Moreover, during HA degradation, calcium and phosphate ions are released into the local environment, inducing an osteogenic response [289]. In addition to bone, HA has also been used for cartilage applications. In one such study, Zhou et al. [290] implanted a composite PGA-HA scaffold seeded with autologous MSCs into a rabbit model, leading to both hyaline cartilage and subchondral bone formation [290]. Another bioceramic used in scaffold construction is bioactive glass. It contains SiO_2_, Na_2_O, CaO, and P_2_O_5_ in specific varying proportions [286]. Further, bioactive glass is able to form bonds with both hard and soft tissue [287]. In one study, it was shown that the addition of 45S5 Bioglass^®^ (Si and Ca ions) in osteoblast cell culture promoted osteoblast proliferation [291]. With regard to skin tissue engineering, the incorporation of 45S5 Bioglass^®^ into PGA meshes increased the scaffold neovascularization both in vitro and in vivo [292]. Muscle regeneration has also been shown to be enhanced where the addition of phosphate bioglass (CaO–Na_2_O–Fe_2_O_3_–P_2_O_5_) supported the attachment, proliferation, and differentiation of muscle precursor cell lines and the formation of myotubes [293].

Additionally, growth factors have also been incorporated into scaffolds to improve function. These soluble proteins, manufactured by cells in the body (e.g., macrophages and fibroblasts) can encourage cell growth, proliferation, differentiation, and migration [286,294,295]. Specific receptor binding on the cell membrane surface elicits a response, which may be critical for tissue regeneration and wound healing [296,297]. Growth factors alone possess a short half-life and are rapidly degraded [298]. However, they can be directly embedded within the scaffold to increase stability, or delivered separately with the scaffold to enhance the overall function. Table 2 summarizes common growth factors used in tissue engineering, along with their scaffold delivery.

## 5. Combined Scaffold Fabrication Techniques

Following the detailed understanding of tissue engineering targets, scaffold fabrication methods, and potential polymers/additives, this review can expand into the combination of multiple fabrication methods to directly target tissue interface. Recently, tissue-engineered scaffold fabrication techniques have been combined to create products with diverse and tunable properties for a wide variety of biomedical functions. These span multiple application types including (i) drug delivery and biomedical tools, (ii) multiple scaffold fabrication methods for a single tissue, and (iii) combined scaffolds targeting multiple tissue types.

### 5.1. Drugs and Biomedical Tools

The combination of scaffold fabrication methods has been used to directly target drug delivery. Transdermal drug delivery systems (TDDS), which have the goal of providing sustained and controlled drug release for a long period of time, have been created using electrospinning and cryogelation. The technique of TDDS administers drugs through the skin and into the bloodstream, therefore avoiding the impact of gastrointestinal metabolism on medicine effectiveness. In a paper by Sa’adon et al. [232], an electrospun PVA layer was combined with a PVA cryogel. The electrospun PVA layer alone possessed high porosity and surface to volume ratio, as well as a structure similar to ECM; however, it absorbed water, was mechanically weak, and the high surface area had the risk of drug overloading the patient. Thus, the combination of electrospun PVA with a PVA cryogel improved the overall mechanical properties. It was demonstrated that increased fiber thickness led to higher tensile strength, and strength was also increased with the number of freeze thaw cycles [232]. Another study found that combining electrospinning and cryogelation resulted in a tough, anisotropic, and chemical crosslinker-free nanofibrous structure composed of PVA and glycerol. The ECM possessed both anisotropic and porous properties, which are crucial for strength and cellular interaction [233,358].

Electrospinning has also been combined with 3D-printing to create multi-layer scaffolds for drug delivery [198]. Electrospun mats of PCL-PGS were deposited onto PCL-PGS 3D-printed scaffolds where the 3D-printed layer provided mechanical support, shape integrity, and bioactive compound release, while the electrospun layer improved mechanical properties and porosity for cell infiltration. This construct was suggested as a tool for tendon/ligament reconstruction due to the increased stiffness provided by the combined scaffold construct [198]. Another study also combined electrospinning and 3D-printed for a dual drug delivery platform in periodontal tissue regeneration [359]. Briefly, a bilayer structure was developed by coating honeycomb-like sheets via electrospinning [359].

Further, greater than two fabrication methods have been combined for biomedical tool development. One such study used a combination of electrospinning, gas foaming, and hydrogel to focus on uncontrolled hemorrhage treatment [199]. PCL electrospun mats were sized as small pellets and expanded using gas foaming. These expanded pieces were then immersed in gelatin hydrogel solution. In this application, the pellets could be injected into a patient via a syringe and, once localized, the structure could be expanded to serve as a synthetic blood clot [199]. This particular study is novel, as the fabrication of injectable and shape adaptable electrospun fibrous materials has previously proved complex.

The fabrication method of 3D-printing has also been combined with gas foaming and particulate leaching. The goal of Liu et al.’s study [360] was to develop a flexible tracheal stent that avoids or reduces complications from mucus plugging, stenosis, and migration, leading to decreased patient discomfort. Indirect 3D-printing was combined with gas foaming and particulate leaching fabrication methods to create 3D-printed chiral and anti-chiral stent frames, filled with a mixture of porogens (65 wt. % NaCl + 35 wt. % NaHCO_3_) and liquid silicone rubber. These porogens were then removed and epithelial cells were seeded onto the stent. Overall, this combined structure stent was shown to be flexible, porous, and biocompatible [360]. 

Finally, diagnostic testing has been explored using multiple tissue fabrication techniques. One study developed a minimally invasive prescreening method for gastric pre-malignancy in the general population [361]. Electrospinning, gas-foaming, and coating were used to develop a construct that could be swallowed and expanded to serve as an absorbable cuboid/sphere structure. Overall, the electrospun nanofiber matrix had a water absorption capacity of 2000–6000% the dry mass, and the expanded sphere/cuboid was able to increase the collection and release of bacteria/viruses, as well as cells, suggesting suitability at collecting samples from the duodenal, gastric, and esophageal tracts [361].

### 5.2. Multiple Scaffold Fabrication Methods for a Single Tissue

Additionally, combination fabrication techniques have be used to develop scaffolds targeting a single tissue type. These examples focus more broadly on improving scaffold characteristics, including cell attachment, by combining different fabrication techniques. One such study combined 3D-printing and electrospinning to produce PLA-based composites for multi-textured biological tissues [229]. A composite was fabricated by blending two layers of 3D-printed material with a central layer of PLA fibers to mimic bone tissue. Further, composite materials including PLA, PLA/PEG, and PLA/PEG with two different amounts of HA were also tested. Similarly, another study also combined 3D-printing and electrospinning to support complex and hierarchical functionalities [362]. 3D-printed PCL scaffolds with 300 µm pore size were directly combined with an electrospun layer of PCL, where multiple mesh layers and fiber densities were investigated. Fiber analysis suggested that the layering of electrospun fibers perpendicularly to the 3D-printed mesh is preferred. Additionally, increasing the number of mesh layers increased cell proliferation, suggesting the ability to tailor 3D-printed and electrospun scaffolds based on fiber alignment, mesh layers, and fiber densities [362].

Electrospinning and hydrogel fabrication have also been combined, resulting in improved cell attachment and mechanical properties. Material concentration, porosity, and pore size were varied, and the composite scaffolds were compared to the hydrogels fabricated without an electrospun layer. Overall, composite scaffolds were able to withstand greater elongation compared to hydrogels fabricated alone, but the Youngs modulus was decreased [191].

Additionally, electrospinning has been combined with salt leaching to fabricate a scaffold that retained both macroporous and nanofibrous characteristics [363]. This combination was targeted as nanofibrous scaffolds provide a large surface area and porous structure; however, as fiber size decreased, cellular infiltration was negatively affected [363]. Similarly, electrospinning and gas foaming techniques have also been combined to treat peripheral nerve injury [364]. Briefly, PLA/silk fibroin nanofiber scaffolds were fabricated as a fill for 3D-printed sponges. It was demonstrated that in these combination scaffolds, porosity was increased compared to the 2D scaffold control [364]. 

In another study, electrospinning, gas-foaming, and hydrogel fabrication techniques were combined, this time to target minimally invasive cell delivery [365]. In this study, expanded nanofiber scaffolds were fabricated by the gas foaming of electrospun scaffolds, where additional gelatin coating resulted in high elasticity. These constructs were created to deliver non-expanded cell-seeded scaffolds in vivo through noninvasive methods, followed by scaffold expansion to repair tissue damage [365].

### 5.3. Combined Scaffolds Targeting Multiple Tissue Types

Tissue engineering research has also combined multiple fabrication techniques to replicate various layers of a single tissue. In one such study [366], 3D-printing, electrospinning, and hydrogel fabrication were combined to create an intervertebral disc (IVD) scaffold targeting IVD degeneration. Anatomically, the IVD consists of the gel-like nucleus pulposus, collagenous annulus fibrosus, and vertebral end-plates. Here, the IVD scaffold framework was produced by 3D-printing PLA and electrospun oriented porous poly(l-lactide)/octa-armed polyhedral oligomeric silsesquioxanes (PLLA/POSS-(PLLA)8) fiber bundles were joined to the construct to simulate the annulus fibrosus. Finally, to replicate the nucleus pulposus, a gellan gum/poly(ethylene glycol) diacrylate (GG/PEGDA) double network hydrogel, seeded with bone marrow mesenchymal stem cells, was also added. Overall, mechanical testing and in vivo analysis demonstrated this combined scaffold as a useful technique in treating IVD [366]. In addition to IVD applications, a bilayer skin scaffold was developed by combining electrospinning (PCL and PVA) and freeze gelation (chitosan and gelatin) [367]. With the addition of multiple scaffold fabrication methods and materials, the tensile strength increased and improved wound healing was noted. This technique has also been used in the vascular tissue engineering realm, where a triple-layer PCL fibrous vascular graft was fabricated by combining E-jet 3D-printing and electrospinning [368]. This vascular graft was composed of three layers with different properties to stimulate appropriate conditions for vascularization. Specifically, the innermost layer was comprised of highly aligned, durable 3D-printed fibers, the middle layer was comprised of dense electrospun fibers, and the outermost layer was comprised of a mixture of electrospun fibers [368]. This multilayered scaffold possessed varying properties, resulting in improved cell growth and infiltration in vivo. Similarly, human heart valve leaflet engineering has also benefited from combination techniques. In a study by Freystetter et al. [369], models of aortic valve cusps were designed using CAD. Conductive PLA was then used to 3D-print the models. To mimic human heart values, three layers of polyurethane were spun onto the models [369]. 

Moving forward, new and future tissue engineering research should combine various fabrication techniques to replicate multiple tissues, focusing on the tissue interface which varies between different tissue types; this is often dictated as the soft to hard tissue interface. Examples of this include bone to tendon, ligament, or cartilage, representing an extreme category of interface design that consists of materials at opposite ends of the physical property spectrum. Soft to hard interfaces are designed to transfer loads between tissues that have very different mechanical properties, which is difficult to replicate both in the lab and in vivo. Currently, healing of a damaged hard to soft tissue interface results in scar tissue that has inappropriate structural properties to replicate the original function; thus, the injured tissues are often subject to repeated damage [72].

## 6. Conclusions and Future Directions

This manuscript focuses on common tissue types, biomaterials, and additives in tissue engineering, providing a guide for navigating future research in tissue interface through the combination of different scaffold fabrication methods. The future of tissue engineering relies heavily on the success of biomaterial scaffolds in supporting appropriate repair, replacement, and regeneration. Despite the increased volume of scaffold fabrication research over the last 30 years, work has primarily focused on single tissue reconstruction—this disregards the body being a dynamic system and the necessity of crosstalk between tissues being highly recognized across healthcare. Thus, there remains a critical gap in the knowledge base that pertains to combining different fabrication methods without disrupting structural integrity and biomaterial properties. This type of research will be significant because the creation of a scaffold unit spanning tissue interface is expected to: (i) improve the fabrication method of combined scaffolds; and (ii) expand current understanding of the effect of biomaterial integration on mechanics. To complete this, the components of tissue-engineered scaffolds, as they relate to specific tissue types, must be explored.

This manuscript first elaborated on the composition, function, and wound healing capacity of commonly targeted tissue in tissue engineering. These tissues included bone, cartilage, ligament, muscle, skin, and tendon. The current goal standard of treatment for each tissue is primarily reconstruction using host tissue (autograft) or donor tissue (allograft). The main disadvantage with these current methods is donor site morbidity and graft failure for autograft and allograft, respectively. Tissue-engineered scaffolds circumvent both problems by providing a matrix adapted for the desired application to facilitate new tissue growth and wound healing. To accomplish this, multiple tissue-engineered fabrication methods can be targeted to tailor scaffold properties to the desired tissue. Various methods of scaffold fabrication were discussed, providing the overall principle, procedure, and advantages/disadvantages of each technique, focused on 3D-printing, bioprinting, electrospinning, gas foaming, hydrogels, cryogels, and solvent casting/particulate leaching. Within each method, there are controllable variables, e.g., applied voltage and flow rate in electrospinning or solvent and particulate type in solvent casting/particulate leaching, that modify the overall scaffold properties. The optimal choice of fabrication method is directly dependent on the necessary parameters and overall application, rending some techniques more suitable than others. Finally, a comprehensive summary of polymers/additives typically used with these tissue engineering scaffold fabrication techniques for specific tissue types were discussed, providing an overview and template for tissue design.

Through a detailed understanding of both tissue composition and the properties of various fabrication methods, future work can harness the strength of multiple fabrication techniques, combining them to improve the current function of scaffolds for different biomedical applications. This review provided information on new and innovative work demonstrating the successful integration of multiple fabrication techniques to improve drug delivery systems, developed multi-scaffold fabrication methods for a single tissue, and combined scaffold fabrication techniques to target multiple tissue types. These methods are vital to the advancement of tissue engineering as a field, as trauma and disease commonly affect more than one tissue type. Due to the distinctive physiological properties of these different tissues, combining multiple fabrication methods can be an effective solution by retaining the advantages from multiple scaffold types. Despite the success in these new studies combining multiple fabrication techniques, a challenge still lies with the creation of tissue interface. Currently, there is limited research on the interaction of different scaffold types to replicate a combination of tissue types, and while tissue-engineered scaffolds show great promise for tissue reconstruction, the interface between scaffold fabrication methods have historically posed a structural challenge due to inappropriate material integration. Thus, it is crucial to gain a better understanding of the physical interaction between tissue-engineered scaffolds, where this review provides the tools to further the development of multi-tissue scaffolds, targeting tissue interface regeneration.

## Figures and Tables

**Figure 1 biomimetics-07-00151-f001:**
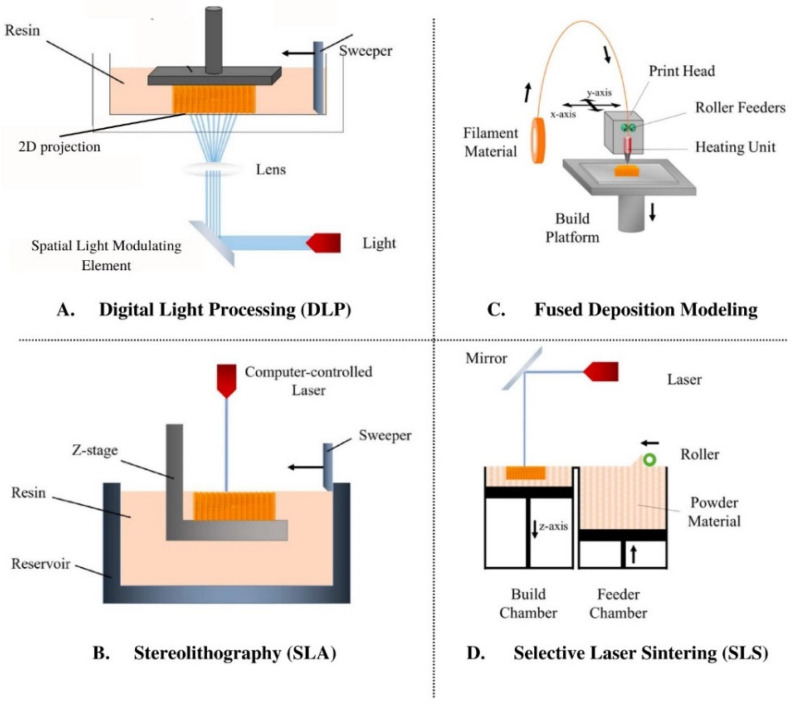
Schematics for four types of 3D-printing: (**A**) Digital Light Processing, (**B**) Stereolithography, (**C**) Fused Deposition Modeling, and (**D**) Selective Laser Sintering. Figure is adapted from Tamay et al., 2019 [81].

**Figure 2 biomimetics-07-00151-f002:**
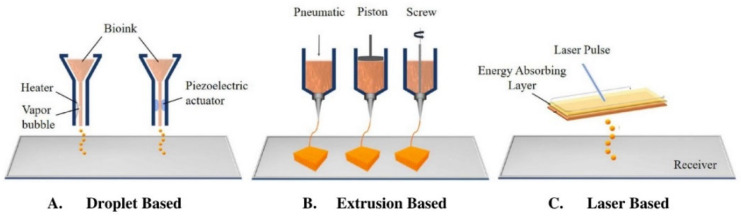
The three types of bioprinting: (**A**) Droplet-based, (**B**) Extrusion-based, and (**C**) Laser-based. Figure is adapted from Tamay et al., 2019 [81].

**Figure 3 biomimetics-07-00151-f003:**
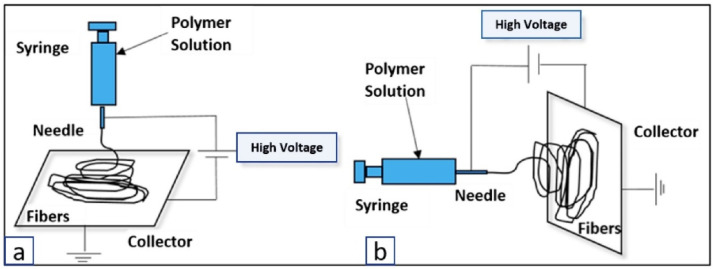
Electrospinning setup for fibrous mesh formation, shown using both (**a**) vertical and (**b**) horizontal spinning [105].

**Figure 4 biomimetics-07-00151-f004:**
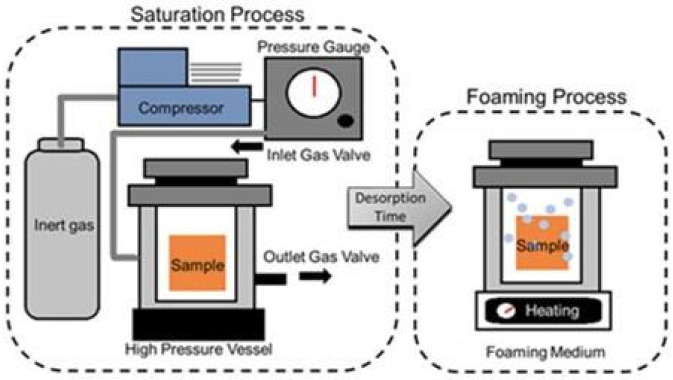
Depiction of the use of a gas foaming agent, high pressure, and high temperature to develop a gas foamed scaffold. The saturation process involves spreading the gas throughout the polymer sample inside a high-pressure vessel or reactor. Once the sample is saturated, it can be placed in a high temperature chamber to allow for desorption. Figure is adapted from Toong et al., 2020 [112].

**Figure 5 biomimetics-07-00151-f005:**
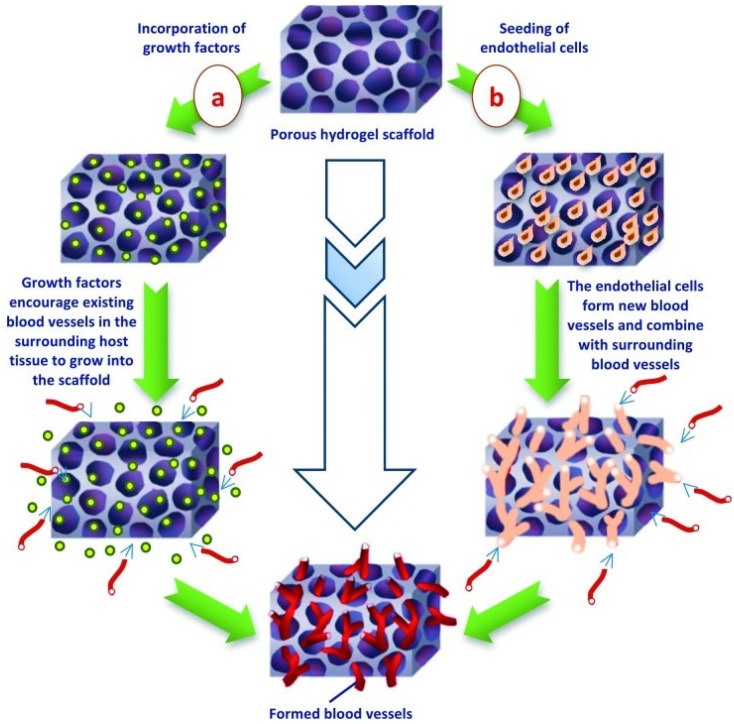
Vascularization of hydrogel scaffold completed by the incorporation of (**a**) regulatory growth factors or (**b**) endothelial cells into the tissue-engineered structure [116].

**Figure 6 biomimetics-07-00151-f006:**
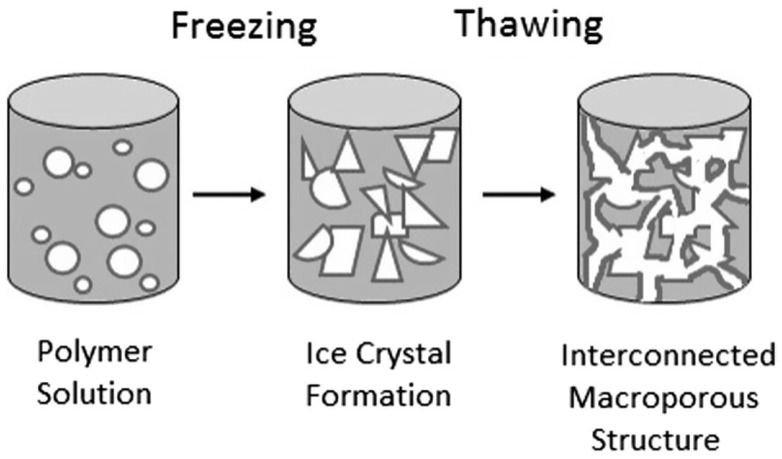
Fabrication of cryogel scaffolds demonstrating the crosslinking and subsequent freezing of the polymer solution, forming a sponge-like, macroporous structure [135].

**Figure 7 biomimetics-07-00151-f007:**
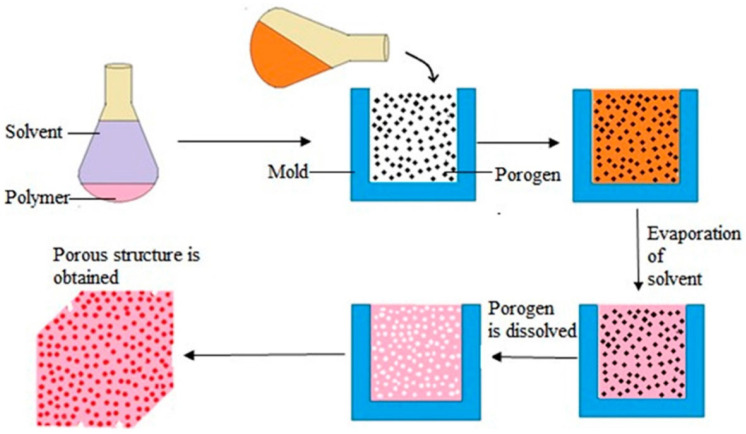
Overview of the solvent casting and particulate leaching process [161].

**Table 1 biomimetics-07-00151-t001:** An overview of commonly used natural, synthetic, and ECM-derived polymers in tissue engineering, along with frequently used fabrication methods, target tissues, and advantages/disadvantages.

Polymer	Strength/Elastic Modulus	Fabrication Method	Target Tissue	Advantage(s)	Disadvantage(s)	Notes:
**Natural Polymers**						
Alginate	Hydrogel: <1 kPa to 1000 kPa [166]	Electrospinning; Hydrogels; Cryogels [167,168]	Bone; Cartilage; Ligament [167]	Biocompatible; Encapsulate cells [167,169,170,171,172].	Lacks mechanical strength; Nondegradable unless ionically crosslinked; Slow degradation [166,167,169,171].	Alginate gels are typically nanoporous (pore size ~5 nm), allowing for rapid diffusion of small molecules through the gel; Potential for drug delivery [167,173].
Chitosan	Fibers: 3500 +/− 780 kPa Scaffold: 70 +/− 10 kPa [174]	Hydrogel; Cryogel [150,175]	Bone; Cartilage; Ligament; Nerves [169,174]	Antimicrobial properties; Biocompatible; Chemically modifiable [169,176,177,178].	Limited cell adhesion; Low mechanical strength [169,176,177,178].	Has been reported to promote bone formation [179].
Collagen I fiber	100–2900 MPa [180]	3D-printing; Electrospinning [181]	Bone; Cartilage; Ligament; Skin; Tendon [181]	Biocompatible; Major component of native ACL [169].	Lacks mechanical strength; Immunogenic [169].	The addition of hyaluronic acid can prolong degradation and serve as a delivery system for chondrocytes in cartilage tissue engineering [182,183].
Gelatin	Scaffold: 10–100 kPa [184,185]	3D-printing; Electrospinning; Hydrogels; Cryogels [150,186,187]	Bone; Cardiac; Ligament; Muscle; Skin; Tendon [188]	Biocompatible; Biodegradable; Cost-effective; Cell compatibility; Low toxicity [189].	No thermal stability; Poor mechanical properties; Short degradation rate [189].	Gelatin is derived from collagen and can be used as a cost-effective substitute [188].
Silk	5–12 GPa for Bombyx Mori with sericin, 15–17 GPa without sericin [190]	Electrospinning; Hydrogel; Cryogel [191,192,193]	Bone; Cartilage; Ligament; Skin [169,174,194]	Good tensile strength [37,169].	Limited cell adhesion; Sericin coating is immunogenic [37,169].	Two main constituents: fibroins and sericin. Sericin, absent in spider silk, acts as a glue for the fibroin fibers and elicits an immune response. These constituents contain (varying) amounts of alanine, glycine, and serine [195].
**Synthetic Materials**						
Poly(caprolactone) (PCL)	0.4 GPa; 3.2 MPa [169,180,196]	3D-printing; Electrospinning; Gas Foaming [197,198,199]	Bone; Ligament; Soft and Hard Tissues [169,200,201,202]	Common FDA-approved suture material; Easily manufactured [169].	Biologically inert; Slow degradation rate (years) [14,169].	Elongation at break 80%, Tg = −60 C, Tm = 60 C [196].
Poly(diaxonane) (PDX)	100,000,000 N/m^2^ = 0.1 GPa, 2–46 MPa [180,203]	Electrospinning [180]	Bone; Cartilage; Ligament [180]	Common FDA-approved suture material; Easily manufactured; Shape memory [169].	Rapid loss of mechanical strength [169].	PDX/50% Hydroxyapatite scaffolds allow for excellent scaffold mineralization for bone tissue engineering [204].
Polyethylene terephthalate (PET)	1.57–5.2 GPa [205]	3D-printing; Electrospinning [206,207]	Bone; Ligament; Tendon [208,209,210]	Biocompatible; Biodegradable; High tensile strength; Stiffness [194,210].	High crystallinity makes it difficult to print [210].	Frequently made into meshes containing allografts/autografts, or meshes for hernia repair [211,212].
Poly(glycolic acid) (PGA)	7.0–10 GPa [213]	Electrospinning; Gas Foaming [214,215]	Cartilage; Skin [216,217]	Common FDA-approved suture material; Easily manufactured [169].	Rapid degradation and loss of mechanical strength; Biologically inert; Acidic degradation byproduct [169].	Frequently used in combination with other materials as coatings (ex. hyaluronic acid) [218,219].
Poly(glycerol sebacate) (PGS)	0.04–1.2 MPa [213,220]	3D-printing; Electrospinning [198,213]	Soft Tissue [201,213]	Biocompatible; Biodegradable; Cost effective; Flexible [213].	Differences between in vivo and in vitro degradation [221].	Fast degradation (6 mo in vitro) [213].
Poly(3-hydroxybutyrate) (PHB)	3 GPa [222]	Electrospinning; Salt Leaching; Solvent Casting [223]	Bone; Cartilage; Skin; Tendon; Nerves [194,223,224]	Biocompatible; Piezoelectric [223].	Brittleness; Hydrophobicity; Low degradation rate [223].	Often combined with 3 hydroxyvaleric acid (HV) to increase degradation rate and reduce crystallinity; elongation at break 2%; Tg = 1–2 C, Tm = 170 [194,196].
Poly(3-hydroxybutyrate-co-3-hydroxy valerate) (PHBV)	Tensile modulus = 1100 MPa [196]	Electrospinning [225]	Cardiac; Cartilage; Liver; Nerve [225]	Biocompatible; Biodegradable; Low toxicity; Piezoelectric; Thermoplasticity [226,227].	Hydrophobic; Low mechanical strength; Often requires additives to promote cell adhesion; Poor mechanical properties [226].	Elongation at break 17%; Tg = 2 C; Tm = 145 C; Tensile strength = 20 MPa [196,225,228].
Poly(lactic-co-glycolic acid) (PLGA)	40.4–134.5 MPa [213]	Electrospinning [197]	Ligament; Vascular [169,197]	Degradation rate can be tailored by changing the ratio of PLA:PGA [14,169].	Acidic degradation byproducts; Biologically inert; Reduce cell adhesion; Non-hydrophobic [14,169].	Degradation rate of 32% weight loss observed at 5 weeks in vitro [213].
Poly(L-lactic acid) (PLLA)	1–4 GPa [213]	3D-printing; Electrospinning [229]	Ligament; Neural; Hard and Soft Tissues [169,230]	Easily manufactured; Improved cell adhesion; Slow degradation rate [169].	Acidic degradation byproduct; Biologically inert [169].	Viscoelastic properties can be improved by using braid-twist method [169].
Polyvinyl alcohol (PVA)	48 +/− 3 GPa [231]	Electrospinning; Cryogels [232,233]	Bone; Skin [234,235]	Biocompatible; Good mechanical properties; Non-toxic [169,231].	Low thermal stability [169,231].	Young’s modulus and compressive strength increases with PVA concentration [236].
**ECM-Derived Polymers**						
Chondroitin sulfate	Hydrogel: 1.2–11.3 kPa [237,238]	Electrospinning; Hydrogel [237,239]	Bone; Cartilage; Neural; Skin [240]	Biocompatible; Biodegradable; Readily available; Water soluble [240].	Differences in material quality; Fast degradation; Low thermal resistance; Tunability Weak mechanical properties [240].	Source: Joint, Nasal, and Tracheal cartilage. Support osteogenesis and suppress bone resorption [241,242,243,244].
Elastin	Bovine ligament: 1 MPa [245]	Electrospinning [246]	Ligament; Skin [247,248]	Provides elasticity, resiliency, cell adhesion and growth; Artificial forms are available [247].	Difficult to purify; May stimulate an immune response [249,250].	Source: Aorta. Used in scaffolding for cell generation therapy; Elasticity provides the necessary mechanical cues for maintaining and expanding hematopoietic stem cells [249,250].
Fibronectin	Fibers: 1–15 MPa [251]	Electrospinning; Hydrogel [249,252]	Bone; Dental Tissue [253]	Promotes cell adhesion, migration, spreading, and proliferation; Wound healing [247,253].	Decreased cell viability; Insufficient cell-anchorage cues [252].	Source: Plasma; Promote cell adhesion and control cellular function via peptide domains. Angiogenic differentiation [249,254,255,256,257].
Heparin/Heparan Sulfate-Derived	Hydrogel: 2.3 kPa [258]	Hydrogel [258]	Bone; Cartilage; Neural; Vascular [258,259,260,261]	Anticoagulant; Anti-inflammatory [127,259,262,263].	Human umbilical vein endothelial cells (HUVEC) metabolic activity is sensitive to heparin (negatively impacted at 10 and 1000 μg/mL) [261].	Excellent matrix for in vitro culture of articular chondrocytes; Facilitate interactions at cell surface receptors [258,264].
Hyaluronic acid	Hydrogel: 200 kPa [265]	Bioprinting; Electrospinning; Hydrogel; Cryogel; Salt Leaching [266,267]	Bone; Cartilage; Skin [268,269,270,271]	Chemically modified (crosslinked) to improve viscoelastic properties [268].	Poor biomechanical properties (viscoelasticity and half-life); Pure hydroxyapatite (HA) gel does not permit adhesion [268,272].	Source: Rooster comb, Umbilical cords, Vitreous humor. The HA-based hydrogel can maintain morphology of chondrocytes [266,267,273,274].
Laminin	Fiber: (blended with PCL) 21.83 kPa	Electrospinning [275]	Nervous System; Skeletal Muscle; Vascular [276]	Biomaterial enrichment; Cell adhesion, differentiation, migration, and wound healing [247].	Difficult to synthesize long peptides and mimic structure [277].	Source: Heart, Placenta. Provide basement membrane-like scaffold. Participate in biological process (angiogenesis and neural differentiation) [249,275,278,279].

**Table 2 biomimetics-07-00151-t002:** Growth factors commonly incorporated in tissue-engineered scaffolds (Adapted from Whitaker, M. et al. [299]).

Growth Factor	Target & Application in Tissue Engineering	Fabrication Methods	References
Bone morphogenetic protein-2 (BMP-2)	Bone regeneration; Cell proliferation	3D-printing; Bioprinting; Electrospinning; Gas foaming; Hydrogel; Cryogel; Solvent Casting/Particulate Leaching	[300,301,302,303,304,305,306,307,308]
Epidermal growth factor (EGF)	Cell proliferation; Neural Stem Cell Differentiation; Wound healing	Bioprinting; Electrospinning; Gas foaming; Hydrogel; Cryogel; Solvent Casting/Particulate Leaching	[309,310,311,312,313,314,315,316,317,318]
Fibroblast growth factor (FGF)	Angiogenesis; Bone/Cartilage Regeneration; Cell Proliferation; Nerve Growth	3D-printing; Electrospinning; Gas Foaming; Solvent Casting/Particulate Leaching	[262,263,314,319,320,321,322,323,324,325,326,327]
Nerve growth factor (NGF)	Axonal Growth; Neurite Extension in Central and Peripheral Nervous Systems	3D-printing; Bioprinting; Electrospinning; Hydrogel; Cryogel	[260,328,329,330,331,332,333,334,335,336]
Platelet-derived growth factor (PDGF)	Angiogenesis; Endothelial Cell Proliferation; Wound Healing	3D-printing; Electrospinning; Gas foaming; Hydrogel	[326,337,338,339,340,341,342,343]
Transforming growth factor-β (TGF-β)	Bone/Cartilage Regeneration; ECM Production	3D-printing; Bioprinting; Electrospinning; Hydrogel; Cryogel; Solvent Casting/Particulate Leaching	[325,326,341,344,345,346,347,348,349,350]
Vascular endothelial growth factor (VEGF)	Angiogenesis; Endothelial Cell Proliferation	3D-printing; Bioprinting; Electrospinning; Gas foaming; Hydrogel; Cryogel; Solvent Casting/Particulate Leaching	[264,303,304,318,351,352,353,354,355,356,357]

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
