# Peer review of "Current Concepts and Methods in Tissue Interface Scaffold Fabrication"

_biomimetics, 2022, doi:10.3390/biomimetics7040151_

Round 1

Reviewer 1 Report

The current paper submitted by Oraya Vesvoranan et al. titled: “Current Concepts and Methods in Tissue Interface Scaffold Fabrication” discusses the different scaffold fabrication techniques and chararistics on multi-tissue damage. The authors had reorganized clear-cut informations and knowledge of tissue engineering for the reader. Despite these, the manuscript lacks of novalty, it's very easy to find similar review papers in biomedical field. The contents of this manuscript could be easily searched and found from other publications. Please find the reference listed below:

1. El-Sherbiny, Ibrahim M., and Magdi H. Yacoub. "Hydrogel scaffolds for tissue engineering: Progress and challenges." Global Cardiology Science and Practice 2013.3 (2013): 38. 

2. Kumar, Abinash, and Anu Jacob. "Techniques in scaffold fabrication process for tissue engineering applications: A review." Journal of Applied Biology and Biotechnology 10.3 (2022): 1-7.

3. Krishna, B. Swathy, and K. Vandana. "Recent Advances in Scaffold Fabrication Techniques for Tissue Engineering." A Holistic and Integrated Approach to Lifestyle Diseases (2022): 251-279.

4. Chimerad, Mohammadreza, et al. "Tissue engineered scaffold fabrication methods for medical applications." International Journal of Polymeric Materials and Polymeric Biomaterials (2022): 1-25.

5. Yoon, Jeong-Yeol. "3D Scaffold Fabrication." Tissue Engineering. Springer, Cham, 2022. 155-174.

Reviewer 2 Report

The authors have prepared a review article and discussed the common tissue types, biomaterials and current tissue scaffold fabrication methods.This review is impressive for the reviewer and would show a significant impact on the tissue engineering and materials science community. The article provides a guide to combining multiple tissue-engineered scaffold fabrication to a product with a variety of functions. However, I have some suggestions before the manuscript to be publish. The title of this article is Current Concepts and Methods in Tissue Interface Scaffold Fabrication, but in the Method part, the authors only describe common scaffold fabrication methods without mentioning which tissue engineering applications are used. From section 1 to section 4, it looks like individual sections are not connected each others and are not related to the title. In my opinion, Table 1 summarised general polymers used in tissue engineering, along with commonly used fabrication methods, target tissues, and pros/cons, which is the core point of this article. Content should be presented in such form. So, I suggest that the authors can reorganise the content to make the manuscript more clearly express the authors’ viewpoint. 

Reviewer 3 Report

Dear authors.

You present a well-detailed review on the concepts and techniques of  scaffold fabrication for tissue regeneration. This review seems to me to be very useful for researchers working on tissue regeneration. I have no corrections to request.

Best regards
